# DIFFUSION SIGFORMER FOR INTERFERENCE TIME-SERIES SIGNAL RECOGNITION

## ABSTRACT

The various interferences in the actual environment make electromagnetic signal recognition challenging, and this topic has extremely important application value. In this paper, a novel interference signal recognition transformer is proposed, named Diffusion SigFormer. Firstly, we explored the interference law of electromagnetic signals and designed a signal interference mechanism. Secondly, diffusion signal denoising modulewas proposed to denoise the input interference signal. We also use various types of noise to improve its denoising effect on electromagnetic signals. Thirdly, SigFormer is designed to extract and classify the denoised signal. For the characteristics of electromagnetic signals, SigFormer leverages 1-D Patch Embedding and combines transformer with convolution. Finally, we conducted experimental verification on datasets RML2016.10a, RML2016.10b and BT dataset. The experimental results show that the proposed method has excellent anti-interference ability.

## 1 INTRODUCTION

Electromagnetic signal recognition is a challenging task in the field of signal processing Sun et al. (2024); Zhai et al. (2024). It is essentially a typical pattern recognition problem, which refers to extracting features from corresponding electromagnetic signals and automatically predicting the category label of the signal. Electromagnetic signal recognition has a wide range of applications in various fields, such as radar Guo et al. (2022); Lang et al. (2021), communication Zhang et al. (2022), human activity recognition Lin et al. (2022); Seong et al. (2024), human expression recognitionChen et al. (2020), etc. This paper specifically studies modulation signals and bluetooth signals.

For electromagnetic signal recognition, analyzing the essential characteristics and mechanisms of signals and achieving electromagnetic signal recognition in complex environments is of great significance. Early electromagnetic signal recognition methods can be generally divided into two categories: likelihood based (LB) methods Dulek (2017); Zheng & Lv (2018); Ramezani-Kebrya et al. (2013); Hameed et al. (2009) and feature-based (FB) methods Majhi et al. (2017); Headley et al. (2008); Alarabi & Alkishriwo (2021); Huang et al. (2016). LB methods use probability theory, hypothesis testing theory, and appropriate decision criteria for electromagnetic signal recognition, while FB methods use feature extraction and classification. In feature extraction, expert systems are used to extract various statistical features of instantaneous amplitude, phase and frequency, such as high-order statistics (HOS) and cyclostationary features. During the classification process, classification algorithms such as Decision Tree, Support Vector Machine (SVM) and Artificial Neural Network (ANN) are designed to identify electromagnetic signals. Although LB methods are optimal in Bayesian estimation, they heavily rely on prior knowledge and parameter estimation. Compared with LB methods, FB methods have stronger robustness and effectiveness, but its recognition performance depends on manually designed features and classifiers.

In recent years, with the continuous development of artificial intelligence, deep learning methods have become a research hotspot in the field of electromagnetic signal recognition. By constructing neural network models, deeper information can be learned from different electromagnetic signal data, improving the precision and efficiency of recognition. MCNet Huynh-The et al. (2020) is a cost-effective Convolutional Neural Network (CNN). It has several specific convolution blocks to simultaneously learn spatiotemporal signal correlations through different asymmetric convolution kernels. MCLDNN Xu et al. (2020) is a three stream Deep Learning framework that integrates

one-dimensional (1-D) convolution, two-dimensional (2-D) convolution, and long short-term memory (LSTM) layers to more effectively extract features from both temporal and spatial perspectives. CGDNet Njoku et al. (2021) is a cost-effective hybrid neural network consisting of shallow convolutional networks, gated recurrent units, and deep neural networks. DAE Ke & Vikalo (2022) based on LSTM denoising autoencoder is designed for automatic modulation recognition (AMC).

However, there are many interference factors such as channel fading and background noise. The electromagnetic environment is becoming increasingly complex, and electromagnetic signals are dynamically unstable. These factors pose significant challenges to signal recognition models. Deep learning models are susceptible to various interferences, leading to a decrease in recognition accuracy. It is crucial to develop an anti-interference electromagnetic signal recognition model to meet practical needs.

In addition, most of the existing deep learning methods for electromagnetic signals are based on CNN LeCun et al. (1998) or RNN Zaremba (2014). Although CNN can extract local features of images and signals well, its indispensable pooling layer will inevitably lead to information loss; RNN (or LSTM Gers et al. (2000), GRU Cho (2014), etc.) can only be calculated unidirectionally. There are two issues with this mechanism. Firstly, the computation of time slices relies on the results of previous time steps, which greatly restricts the parallelism of the model. Secondly, prior information is lost during the sequential calculation process. Although gate structures have been proposed, they only partially alleviate the loss of long-term dependencies. The self-attention mechanism in Transformer Vaswani (2017) can effectively solve the above problems. It can not only extract global contextual information, but also has strong parallelism.

Recently, diffusion models have shown great potential in the field of image generation. The diffusion model has several advantages over other generative models, such as Generative Adversarial Networks (GANs) Goodfellow et al. (2014); Mirza & Osindero (2014); Radford (2015); Zhu et al. (2017). They are easier to train and do not experience issues such as pattern crashes or low-quality output. However, electromagnetic signal denoising based on diffusion models has not been fully explored.

In this paper, inspired by the powerful denoising ability of diffusion models and considering the interference of a large number of signals in reality, combined with the powerful temporal processing capability of transformers, we propose a novel signal recognition method based on diffusion model and transformer. The main contribution of this paper can be summarized as follows:

1) An anti-interference signal recognition method named Diffusion SigFormer is proposed. It consists of Diffusion signal denoising module (DSDM) and classification model SigFormer.

2) A mechanism for electromagnetic signal interference is designed. This mechanism is used to add an appropriate amount of interference to clean signals, facilitating subsequent research on anti-interference electromagnetic signal recognition.

3) In DSDM, we fix the time $t$ and improve its denoising effect on electromagnetic signals by changing the interference rate and type of noise.

4) The basic block of SigFormer combines convolution and Transformer, which enables the model to have excellent local feature extraction ability and global context modeling ability.

## 2 METHOD

In this section, we first present the overall architecture of the proposed method. Secondly, the signal interference mechanism is introduced. Thirdly, the structure of diffusion signal denoising module and its denoising process are expounded. Finally, we provide a detailed introduction to the SigFormer.

### 2.1 OVERALL ARCHITECTURE

The overall framework of Diffusion SigFormer is shown as Fig. 1. It is generally divided into three parts: signal interference module, diffusion signal denosing module and SigFormer. The backbone of the diffusion model is the encoder-decoder structure Ronneberger et al. (2015), and SigFormer consists of several SigFormer Blocks and a classifier.

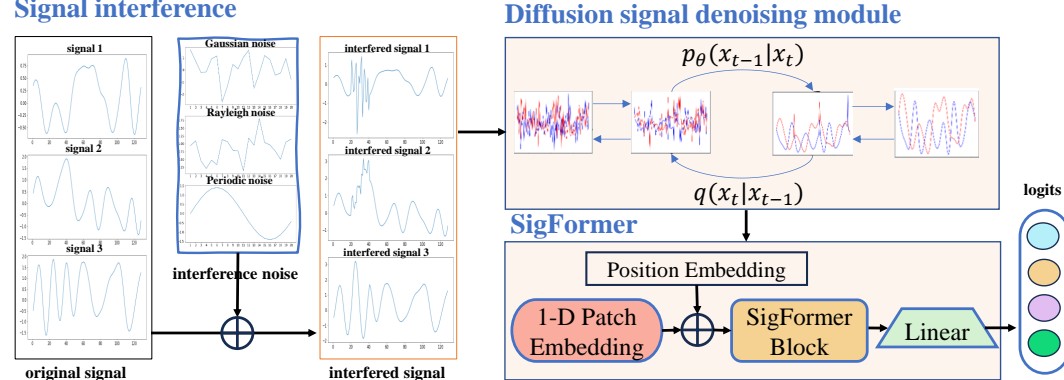

Figure 1: Overall framework of our proposed Diffusion SigFormer

Specifically, the interfered signal is first input into the diffusion model, which denoises it and outputs the clean signal as the input to the SigFormer. Next, SigFormer performs 1-D Patch Embedding on the interfered signal, converts the continuous signal into a series of tokens, adds positional encoding, and adds a class token for classification. These tokens are then input into SigFormer Block for feature extraction. It is worth noting that only the class token is used for the final prediction.

## 2.2 SIGNAL INTERFERENCE MECHANISM

Intuitively, the amplitude of noise should match certain dimensions of the signal itself, and neither too large nor too small can effectively distinguish the changes in signal recognition accuracy caused by noise interference. We define signal interference rate (SIR) as the ratio of the root mean square amplitude of the interference noise to the root mean square amplitude of the original electromagnetic signal, as shown in follows.

$$SIR = \frac{\gamma}{\sqrt{\sum_i A^2_{signal_i}}} \tag{1}$$

where $\gamma$ is the coefficient of noise. For the interference process, We first calculate the root mean square amplitude of the original signal, take the disturbance rate value and multiply it by the amplitude to obtain the magnification of the added noise. After multiplying the unit noise by the magnification, we add it to the original signal to obtain the interference signal.

## 2.3 DIFFUSION SIGNAL DENOISING MODULE

The diffusion model has strong physical properties, as well as natural adaptability and affinity for physical signals. Here, we applied it to the field of electromagnetic signals, and its structure is shown in Fig. 2. It gradually adds noise to clean signals, and as the amount of noise increases, the signal becomes increasingly chaotic. When the amount of noise is large enough, the original clean signal becomes almost pure noise. During the process of adding noise, use neural networks such as U-net to predict the amount of noise added. When denoising, a noisy signal is input and the neural network gradually denoises it to obtain a clean signal.

Specifically, DSDM considers the process of adding noise as a Markov process. Given the condition of $x_0$, by modeling the joint distribution from $x_1$ to $x_t$, the entire process can be modeled. Combining Markov properties, the following formula can be obtained.

$$q\left(x_{1:T}|x_0\right) := \prod_{t=1}^{T} q\left(x_t|x_{t-1}\right) \tag{2}$$

where $q\left(x_t|x_{t-1}\right) = N\left(x_t; \sqrt{1-\beta_t}x_{t-1}, \sqrt{\beta_t}I\right)$. It can be seen that each state node in the Markov chain follows a Gaussian distribution and is only related to the previous state, that is, the expression of the current distribution is determined by the previous observation. If the current distribution is known, the value of the current sample can be obtained using reparameterization techniques, as

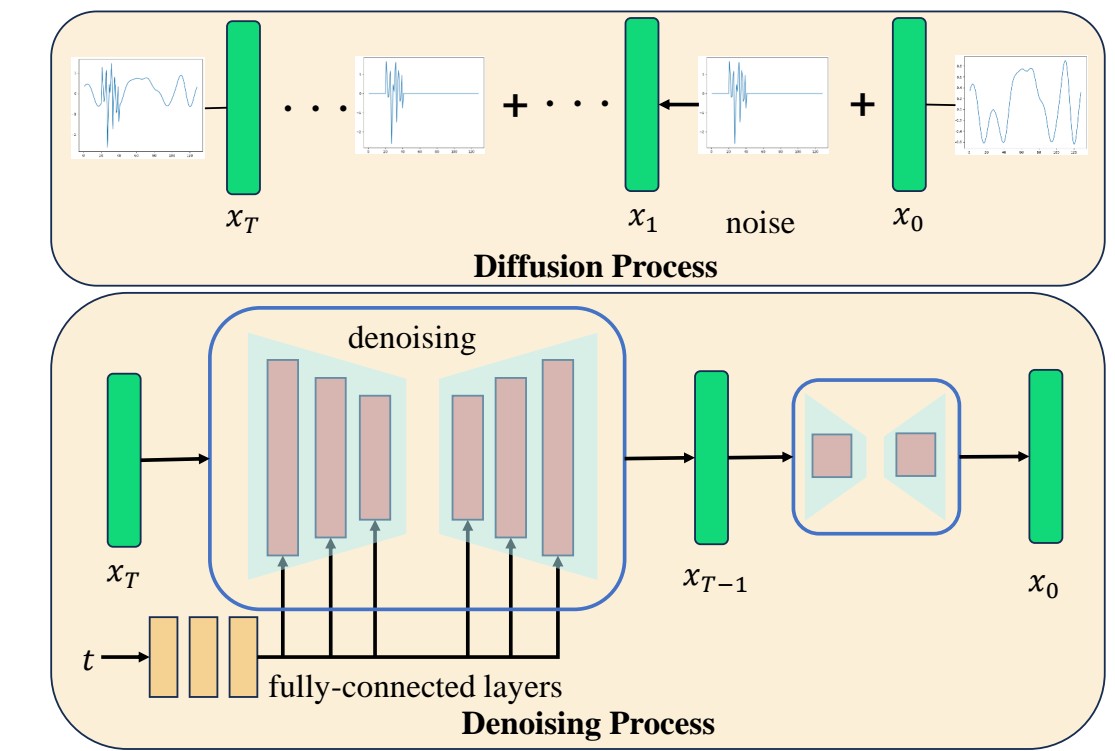

Figure 2: The framework of DSDM

shown in Formula 3.

$$x_t = \sqrt{\alpha_t} x_{t-1} + \sqrt{1 - \alpha_t} \epsilon \tag{3}$$

where $\alpha_t = 1 - \beta_t$, it is a customizable hyperparameter, $\epsilon \sim N(0, I)$. There are currently two main ways of defining sequences, namely linear sequences and cosine sequences.

In DSDM, reparameterization can better represent the recursive relationship between random variables. After reparameterization, $x_T$ can be recursively derived from the original input $x_0$, as shown in Formula 4.

$$x_T = \sqrt{\bar{\alpha}_T} x_0 + \sqrt{1 - \bar{\alpha}_T} \epsilon \tag{4}$$

where $\bar{\alpha}_T = \prod_{t=1}^{T} \alpha_t$. The denoising process involves iterating Gaussian noise $x_T$ step by step back to $x_0$. For the true inverse transfer distribution function, Formula 5 can be obtained using Bayesian formula.

$$q(x_{t-1}|x_t) = \frac{q(x_t|x_{t-1}) q(x_{t-1})}{q(x_t)} \tag{5}$$

Due to the unknown initial distribution, it is not possible to directly obtain the edge distribution from the joint distribution. However, given the transition probability distribution of each state and the properties of Markov chains, Formula 6 can be obtained.

$$q(x_{t-1}|x_t, x_0) = \frac{q(x_t|x_{t-1}) q(x_{t-1}|x_0)}{q(x_t|x_0)} \tag{6}$$

After further reparameterization, Formula 7 can be obtained.

$$x_{t-1} = \frac{1}{\sqrt{\alpha_t}} \left( x_t - \frac{1 - \alpha_t}{\sqrt{1 - \bar{\alpha}_t}} \epsilon \right) + \sigma_t z \tag{7}$$

Note that the noise $\epsilon$ is unknown.

Fitting the distribution of $x_0$ is essentially a problem of using probability models to solve the optimal parameter estimation, and the most classic method is maximum likelihood estimation. So the

optimization is to minimize the negative logarithmic likelihood of the sample: $-\log(p_\theta(x_0))$, it is equivalent to minimizing $KL[q(x_{t-1}|x_t, x_0)|p_\theta(x_{t-1}|x_t)]$, Simply, it equals Formula 8.

$$\left\| \epsilon - \epsilon_\theta\left(\sqrt{\bar{\alpha}_t}x_0 + \sqrt{1 - \bar{\alpha}_t}\epsilon, t\right) \right\|^2 \tag{8}$$

In this way, the problem is transformed into an optimization problem for the noise predictor.

In the forward denoising process, according to Formula 4, the coefficient ratio of noise to signal will vary with the change of t. In order to satisfy the interference rate relationship we define, we fix the time t and set the coefficients of both to $\sqrt{0.5}$, so that the coefficient ratio of the two is 1. In this way, to set different interference rates, we only need to change the coefficient of the noise $\epsilon$. The algorithm flow for training and inference of DSDM is shown in A.3.

## 2.4 SIGFORMER

Transformer has attracted great attention and has held a dominant position since it was proposed. It is known that Transformer was originally used in the field of NLP. Its unique self-attention mechanism makes it perform well in context dependent tasks. In order to transfer it to the image domain, ViT divides the image into several patches, and encodes each patch into a vector, so that an image is encoded into a sequence. At the same time, in order to preserve the spatial relative position information in the image, ViT adds position encoding to each patch, so that ViT can better extract features from the image.

Electromagnetic signals also satisfy context dependent relationships. Inspired by the ViT concept, we proposed SigFormer. Specifically, we also divide the signal into several patches. However, unlike images, signals are one-dimensional data, so we perform 1-D Patch Embedding on them. Similarly, we add positional encoding to it to preserve the temporal information in the signal. After Patch Embedding, signals are converted into a series of tokens. At this point, we concatenate these tokens with a predefined class token and input them to the encoder. Note that only the class token are used for the final classification. The linear layer takes class tokens as input and outputs the predicted probabilities for each category.

In addition, we found that when the Transformer encoder is directly used to extract signal features, the classification accuracy is average and the training is unstable. Therefore, we have made modifications to the Transformer's encoder. Specifically, we added residual convolution between Attention and MLP to enable the model to better focus on local fine-grained features without losing global contextual information, while making training more stable. We name the modified module SigFormer Block, and its structure is shown in Fig. 3.

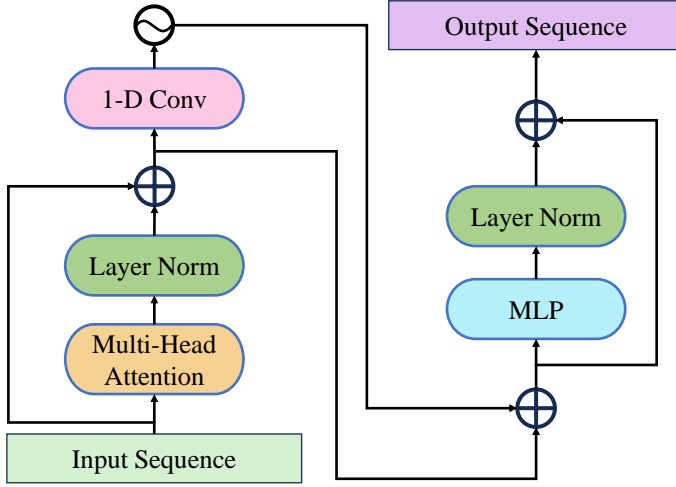

Figure 3: The details of SigFormer Block

Set input $X \in \mathbb{R}^{C \times L}$, the forward propagation process is shown in Formula 9.

$$X = Attention(Norm(X)) + X$$
$$X = Act(Conv(X)) + X \quad (9)$$
$$X = MLP(Norm(X)) + X$$

where Norm is Layer Normalization, and Act is ReLU activation function.

### 2.5 LOSS FUNCTION

#### 2.5.1 DENOISING LOSS

The process of training DSDM is essentially to reduce the distribution difference between predicted noise and real noise. The mean squared error (MSE) loss function is used to optimize this process, as shown in follows.

$$MSE = \frac{1}{2N} \|\epsilon - \hat{\epsilon}\|^2 \quad (10)$$

where $\epsilon$ is real noise, $\hat{\epsilon}$ is predicted noise.

#### 2.5.2 CLASSIFICATION LOSS

The cross entropy loss function is used to optimize SigFormer, as shown in Formula 11.

$$CrossEntropy = -\frac{1}{N} \sum_{i=1}^{N} \sum_{K=0}^{K-1} y_{ik} \log \hat{y}_{ik} \quad (11)$$

where $N$ is the number of samples, $y_{ik}$ is the true label of the i-th sample belonging to the k-th category, and $\hat{y}_{ik}$ is the predicted probability that the i-th sample belongs to the k-th category.

## 3 EXPERIMENT

### 3.1 DATASET

The RML2016.10a dataset is a widely used modulation recognition dataset in the field of wireless communication. The dataset consists of 11 modulation modes, including 8 digital modulation modes and 3 analog modulation modes. For digital modulation, the Shakespearean ASCII format was used. For analog modulation, a continuous speech signal is used as the data source, mainly composed of some raw speech with off time.

The RML2016.10b dataset has a larger scale compared to the RML2016.10a dataset, which has 1200000 samples and includes 8 digital modulation modes and 2 analog modulation modes. The data source also comes from Shakespeare's works and TV series. .

BT dataset is a large-scale Bluetooth signal dataset. It is collected by two data acquisition systems. The first acquisition system collected Bluetooth signals from different smartphone devices at three sampling rates of 5, 10, and 20Gsps, while the second acquisition system collected Bluetooth signals at a sampling rate of 250Msps. To collect Bluetooth signal data, a total of 27 different smartphones were used. This dataset collected Bluetooth signal data from 86 smartphones, with each device recording 150 samples and a total of approximately 12900 records. More detailed information of the three datasets can be seen in A.4.

### 3.2 EVALUATION METRICS

For the above three datasets, we use precision as our evaluation metric, which is defined as follows.

$$precision = \frac{TP}{TP + FP} \quad (12)$$

where TP represents the number of correctly predicted positive samples, and FP represents the number of incorrectly predicted positive samples.

### 3.3 IMPLEMENTATION DETAILS

The Adam algorithm is used to train DSDM and SigFormer. For DSDM, we set the learning rate to 2e-4. For SigFormer, the learning rate is set to 1e-5. For RML2016.10a and RML2016.10b, we trained on data with a signal-to-noise ratio of 18dB and set the batch size to 500. For BT, we trained on data with a sampling rate of 5 and set the batch size to 25. To maintain fairness and consistency, we divided the training and validation sets in a ratio of 4:1. We trained using the Pytorch framework on $1\times$ NVIDIA 3090 GPU.

For signal interference, we investigated various types of noise present in electromagnetic signal environments and studied the characteristics of the three most common types of noise, namely Gaussian noise, Rayleigh noise, and Periodic noise. The characteristic of Gaussian noise is that it has a constant power spectral density at all frequencies, resulting in equal capacity random fluctuations at different frequencies. Rayleigh noise has the characteristics of randomness and irregularity. The distribution of its noise values in time and space is random, and the amplitude is irregular. Periodic noise is a spatial domain noise related to a specific frequency, typically manifested as pulse pairs of sine waves in an image.

It is found that when unit noise is directly added to all feature points, the recognizability of the signal significantly decreases. In order to explore a suitable interference scale that can effectively distinguish changes in accuracy, we took the interference length as the independent variable and conducted experiments on RML2016.10a dataset with a series of different values. We also plotted time-domain waveform diagrams corresponding to the interference length, as shown in Fig.4. We ultimately chose a interference length of 20. When interfering BT dataset with noise, considering its large feature dimension, we selected 50 consecutive feature points for noise interference.

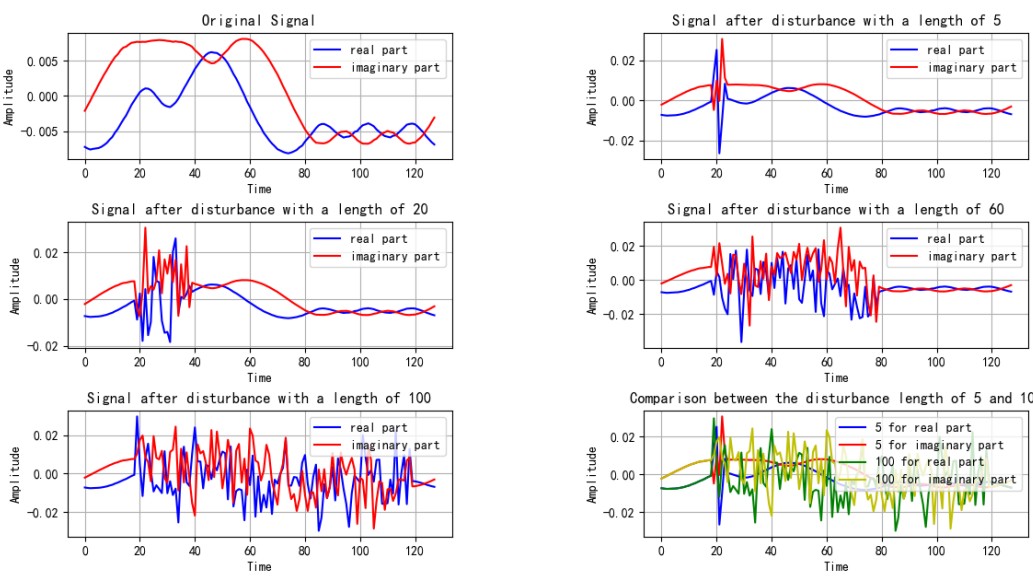

Figure 4: Time-domain waveform diagrams of interference with various length

We conducted signal interference experiments with interference rates ranging from 1 to 10 integers.The interference effect is visualized in Fig.5.

### 3.4 EXPERIMENTS ON RML2016.10A

On RML2016.10a dataset, we conducted comparative experiments between our method and some classic deep learning methods, and the results are shown in Table 1. For ViT and Mamba Gu & Dao (2023), we used the same preprocessing method as SigFormer, which is to use 1-D convolution for Patch Embedding, divide the signal into several patches, and add positional encoding. As can be seen, our method has the highest recognition accuracy. Compared to CNN-based methods such as CGDNet and DCNNPF, SigFormer demonstrates significant advantages, highlighting

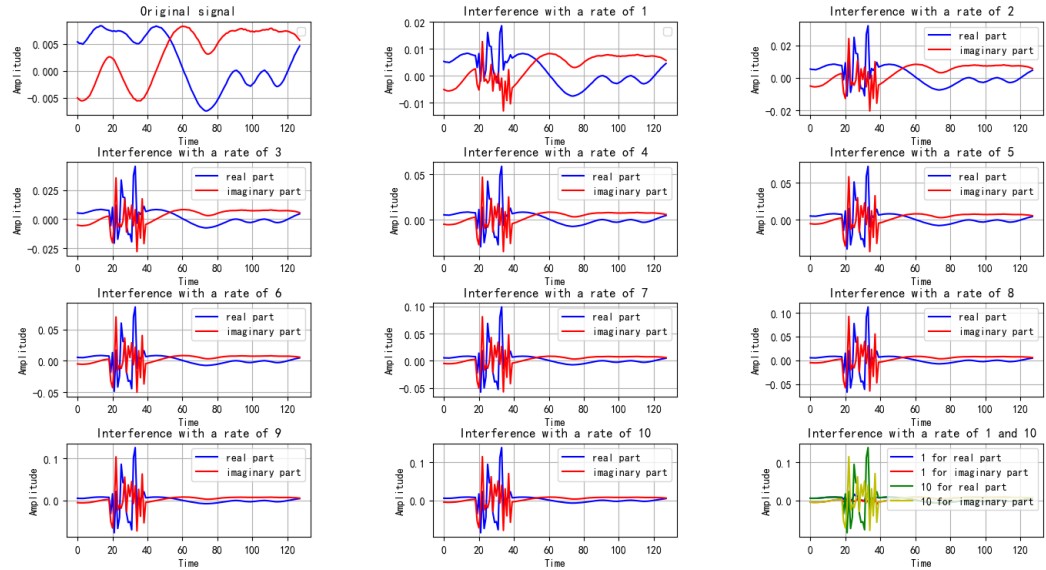

Figure 5: Time-domain waveform diagrams of interference with various rate

Table 1: Results without noise interference on RML2016.10a dataset

| Model | 8PSK | AM-DSB | AM-SSB | BPSK | CPFSK | GFSK | PAM4 | QAM16 | QAM64 | QPSK | WBFM | total |
|---|---|---|---|---|---|---|---|---|---|---|---|---|
| CGDNet | **0.894** | 0 | 0.951 | 0.662 | 0.971 | 0.958 | 0.817 | 0.412 | 0.094 | 0.015 | **0.914** | 0.541 |
| DCNNPF | 0 | 0.922 | 0.016 | 0.088 | 0 | 0 | 0.107 | 0.646 | 0.264 | 0 | 0.36 | 0.263 |
| LSTM | 0.186 | 0.959 | 0.736 | 0.319 | 0.377 | 0.947 | 0.482 | 0.793 | 0.818 | 0.255 | 0.36 | 0.607 |
| ViT | 0.824 | 0.831 | **0.989** | 0.966 | 0.981 | 0.942 | 0.97 | 0.551 | 0.71 | **0.852** | 0.518 | 0.798 |
| Mamba | 0.484 | 0.744 | 0.951 | 0.946 | **0.99** | **0.984** | 0.914 | 0.879 | 0.896 | 0.520 | 0.624 | 0.825 |
| SigFormer | 0.761 | **0.991** | 0.973 | **0.980** | 0.986 | 0.953 | **0.975** | **0.891** | **0.927** | 0.821 | 0.381 | **0.883** |

the importance of global contextual information for electromagnetic signal modulation recognition. Compared to Vision Transformer, SigFormer has an accuracy 10% higher, demonstrating the effectiveness of combining convolution and attention. In addition, the accuracy of SigFormer is also higher than that of the new architecture Mamba, and the potential of Mamba in the field of signal recognition still needs to be explored.

We used Gaussian noise, Rayleigh noise, and periodic noise to interfere the data in RML2016.10a dataset, and the identification results are summarized in Table 2. We randomly select 20 consecutive feature points and interfere all samples with noise. It can be seen that as the interference rate increases, the recognition precision of the model gradually decreases, and the magnitude of the decrease becomes smaller. We use the above-mentioned noise for training DSDM. It can be seen that the accuracy of denoising and recognition remains at a high level, and is very close to the accuracy without noise interference.

Table 2: Results of noise interference on RML2016.10a dataset

| Noise type | Denoising | SIR | | | | | | | | | |
|---|---|---|---|---|---|---|---|---|---|---|---|
| | | 1 | 2 | 3 | 4 | 5 | 6 | 7 | 8 | 9 | 10 |
| Gaussian | No | 0.707 | 0.620 | 0.595 | 0.572 | 0.560 | 0.557 | 0.555 | 0.551 | 0.549 | 0.546 |
| | Yes | 0.853 | 0.860 | 0.858 | 0.857 | 0.853 | 0.852 | 0.862 | 0.854 | 0.855 | 0.851 |
| Rayleigh | No | 0.709 | 0.690 | 0.677 | 0.662 | 0.649 | 0.641 | 0.635 | 0.632 | 0.629 | 0.623 |
| | Yes | 0.853 | 0.860 | 0.858 | 0.857 | 0.853 | 0.852 | 0.862 | 0.854 | 0.855 | 0.851 |
| Periodic | No | 0.648 | 0.614 | 0.588 | 0.582 | 0.582 | 0.582 | 0.575 | 0.566 | 0.563 | 0.562 |
| | Yes | 0.838 | 0.846 | 0.849 | 0.847 | 0.852 | 0.845 | 0.845 | 0.842 | 0.845 | 0.837 |

Table 3: Results without noise interference on RML2016.10b dataset

| Model | 8PSK | AM-DSB | BPSK | CPFSK | GFSK | PAM4 | QAM16 | QAM64 | QPSK | WBFM | total |
|-------|------|--------|------|-------|------|------|-------|-------|------|------|-------|
| CGDNet | 0.502 | 0.976 | 0.731 | 1 | 0.738 | 0.794 | 0.022 | 0.648 | 0.023 | 0 | 0.507 |
| DCNNPF | 0 | 1 | 0 | 0 | 0 | 0 | 0.209 | 0.826 | 0 | 0.335 | 0.285 |
| LSTM | 0 | 0 | 0 | 0 | 0 | 0 | **0.985** | 0.015 | 0 | 0 | 0.167 |
| ViT | 0.919 | 1 | 0.98 | 0.994 | 0.982 | **0.988** | 0.587 | 0.594 | 0.934 | 0.331 | 0.789 |
| Mamba | **0.931** | 1 | 0.983 | 1 | **1** | 0.976 | 0.888 | **0.891** | 0.937 | **0.42** | **0.899** |
| SigFormer | 0.929 | 1 | **0.988** | 0.998 | 0.995 | 0.985 | 0.875 | 0.859 | **0.951** | 0.415 | 0.892 |

Table 4: Results of noise interference on RML2016.10b dataset

| Noise type | Denoising | SIR | | | | | | | | | |
|------------|-----------|-----|-----|-----|-----|-----|-----|-----|-----|-----|-----|
| | | 1 | 2 | 3 | 4 | 5 | 6 | 7 | 8 | 9 | 10 |
| Gaussian | No | 0.590 | 0.450 | 0.422 | 0.414 | 0.406 | 0.400 | 0.396 | 0.387 | 0.387 | 0.384 |
| | Yes | 0.886 | 0.885 | 0.882 | 0.882 | 0.885 | 0.883 | 0.886 | 0.882 | 0.881 | 0.880 |
| Rayleigh | No | 0.465 | 0.478 | 0.475 | 0.495 | 0.481 | 0.473 | 0.483 | 0.465 | 0.451 | 0.460 |
| | Yes | 0.888 | 0.890 | 0.890 | 0.891 | 0.888 | 0.889 | 0.889 | 0.885 | 0.885 | 0.884 |
| Periodic | No | 0.393 | 0.383 | 0.382 | 0.377 | 0.388 | 0.377 | 0.367 | 0.372 | 0.367 | 0.362 |
| | Yes | 0.885 | 0.885 | 0.884 | 0.884 | 0.886 | 0.885 | 0.885 | 0.883 | 0.882 | 0.882 |

## 3.5 EXPERIMENTS ON RML2016.10B

Table 3 summarizes the recognition precision of the comparative algorithms under the condition of no noise interference on RML2016.10b dataset. Table 4 list the accuracy of three types of noise interferences on RML2016.10b dataset. It can be seen that the experimental results on RML2016.10b and RML2016.10a are similar. In the noise free interference comparison experiment, although Sig-Former's accuracy is not the highest, it is very close to the highest accuracy, and the accuracy is even higher on RML2016.10b. In the noise interference experiment, the three types of noise have a greater impact on accuracy, but the denoising effect of the diffusion model is still very good. Under all interference rate conditions, the recognition accuracy after denoising is very close to the accuracy without noise interference.

## 3.6 EXPERIMENTS ON BT

Tables 5 and 6 summarize the experimental results on the BT dataset. In the noise free interference comparison experiment, SigFormer still achieved the highest accuracy. In the noise interference experiment, unlike before, considering the large dimensionality of the BT dataset, we interfered 50 consecutive feature points. As can be seen, the experimental results are similar to the previous two datasets. For three different types of noise, under all interference rate conditions, the accuracy after denoising is very close to the accuracy without noise interference.

## 3.7 VISUALIZATION ANALYSIS

Confusion matrix is a visual tool used to display the difference between predicted and actual values by category. All classification results of all classes are displayed in a confusion matrix, where each column represents the predicted modulation mode and each row represents the true modulation

Table 5: Results without noise interference on BT dataset

| Model | Iphone | LG | Samsung | Sony | Total |
|-------|--------|------|---------|------|-------|
| CGDNet | 0.802 | 0 | 0.205 | 0.655 | 0.541 |
| DCNNPF | **1** | 0 | 0 | | 0.516 |
| LSTM | 0.966 | **0.894** | **0.845** | 0.825 | 0.823 |
| Transformer | 0.787 | 0.263 | 0.409 | 0.983 | 0.653 |
| Mamba | 0.767 | 0.164 | 0.451 | 0.879 | 0.639 |
| SigFormer | 0.909 | 0.544 | 0.697 | **1** | **0.824** |

Table 6: Results of noise interference on BT dataset

| Noise type | Denoising | SIR | | | | | | | | | |
|---|---|---|---|---|---|---|---|---|---|---|---|
| | | 1 | 2 | 3 | 4 | 5 | 6 | 7 | 8 | 9 | 10 |
| Gaussian | No | 0.825 | 0.767 | 0.702 | 0.692 | 0.671 | 0.665 | 0.661 | 0.667 | 0.659 | 0.667 |
| | Yes | 0.825 | 0.825 | 0.824 | 0.824 | 0.827 | 0.825 | 0.825 | 0.827 | 0.827 | 0.822 |
| Rayleigh | No | 0.824 | 0.804 | 0.776 | 0.725 | 0.698 | 0.696 | 0.686 | 0.688 | 0.696 | 0.686 |
| | Yes | 0.829 | 0.829 | 0.824 | 0.825 | 0.827 | 0.822 | 0.827 | 0.825 | 0.829 | 0.827 |
| Periodic | No | 0.804 | 0.782 | 0.765 | 0.749 | 0.7 | 0.671 | 0.653 | 0.663 | 0.659 | 0.661 |
| | Yes | 0.825 | 0.824 | 0.827 | 0.825 | 0.824 | 0.825 | 0.827 | 0.822 | 0.820 | 0.822 |

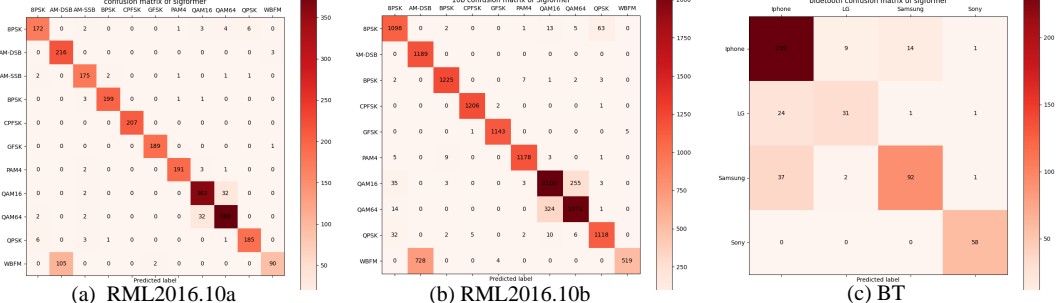

    (a) RML2016.10a           (b) RML2016.10b           (c) BT

Figure 6: Confusion matrix of SigFormer without noise.

mode. The values on each grid represent the predicted probability of the corresponding modulation mode. The visualization results without interference on RML2016.10a, RML2016.10b and BT are shown in Fig.6 On RML2016.10a, it can be seen that SigFormer has a low recognition accuracy for WBFM, and WBFM is more commonly identified as AM-DSB. By plotting the time-domain waveform diagrams of the two, we found that WBFM and AM-DSB have a high similarity in waveform, both of which are close to horizontal lines, making it difficult to distinguish them.The same situation can also be observed in the experiment on RML2016.10b dataset. On BT, the error rate of SigFormer is higher, which may be related to the large dimensionality of Bluetooth signal features and weak waveform regularity.

Overall, the experimental results on RML2016.10a, RML2016.10b, and BT validated the high accuracy and robustness of the diffusion SigFormer. Firstly, SigFormer has demonstrated excellent recognition ability in noise free experiments, with the highest accuracy on both the RML2016.10a and BT, and very close to the highest accuracy on RML2016.10b dataset. For various types of noise, under all interference rate conditions, the diffusion SigFormer can achieve an accuracy very close to that of noise free interferences.

## 4 CONCLUSION

In this paper, we propose a novel method named Diffusion SigFormer for identifying interfered electromagnetic signals, aiming to solve the problem of complex and easily interfered electromagnetic signals in practical environments. Initially, a signal interference mechanism is designed to add interference to the original signal. Then diffusion signal denoising module is designed to denoise the interfered signal. Finally the denoised signal is input to the SigFormer for recognition. SigFormer combines Transformer with convolution, enabling the model to have excellent local feature extraction capabilities and perform well in global context modeling. The experimental results show that our method not only has high accuracy without noise interference, but also approaches the accuracy without noise interference in various types of noise interferences.

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

# A RELATED WORKS

## A.1 DIFFUSION MODEL FOR SIGNAL DENOISING

Diffusion models were originally used for image generation, and compared to other generation models, they are easier to train and can generate more diverse images Ho et al. (2020); Song et al. (2020); Dhariwal & Nichol (2021); Ho & Salimans (2022). Recently, diffusion models have been increasingly applied in the field of signal denoising. In Lan & Huang (2024), a diffusion probability model is adopted to diffuse and reverse process seismic signals, simulate noise pollution and removal processes, and achieve effective signal recovery under different noise conditions. It overlays actual noise and Gaussian noise as synthetic noise in the forward diffusion process, and extracts the time-frequency characteristics of the resulting noise seismic signal as a conditional aid for model training. In Zhu et al. (2023), the diffusion model is applied to denoise Distributed Acoustic Sensing (DAS) vertical seismic profile (VSP) data by first training the diffusion model on a new synthetic dataset that adapts to changes in acquisition parameters. The trained model is used to suppress noise in synthesized and field DAS-VSP data. Deng et al. (2024) proposes a constrained variational diffusion model (VDM) that extends the original VDM by combining constraint methods. By introducing constraint conditions to control the generation process of the model. It receives noisy signals as input and generates specific denoised signals through constrained VDM.

## A.2 ELECTROMAGNETIC SIGNAL RECOGNITION BASED ON TRANSFORMER

Transformer uses self-attention to model global contextual information, which first demonstrated strong performance advantages in the field of natural language processing (NLP) Vaswani (2017); Xiong et al. (2020); Devlin (2018); Radford (2018), and later successfully migrated to the field of computer vision. Representative models include Vision Transformer Dosovitskiy (2020) and Swin Transformer Liu et al. (2021), which have become mainstream models in the field of Computer Vision.

In recent years, Transformers have been increasingly applied in the field of signal processing, including in the direction of electromagnetic signal modulation recognition. In Cai et al. (2022), Transformer was first applied to AMC problem. Transformer combines the global information of each sample sequence and uses semantically relevant information for classification. In Zheng et al. (2022), an improved Transformer modulation recognition model based on GLU was proposed, which combines the advantages of CNN's efficient parallel operation and RNN's ability to fully extract global information of temporal signal context. In Li et al. (2022), a wireless signal modulation pattern recognition method based on Transformer was proposed. This method first segments the data using a fixed size window. Then, the segmented data is projected onto a vector sequence and input into the Transformer module to model and mine the relationship between the signal waveform and modulation mode. ResSwinT–SwinT Ren et al. (2023) is a two-component signal recognition framework. It converts the normalized grayscale time–frequency images (TFIs) of radar signals into a Swin Transformer feature extraction network (SwinT), and the Residual Swin Transformer Denoising Network (ResSwinT) is initialized with low signal-to-noise ratio predictions from a signal-to-noise ratio classifier to reconstruct a clean TFI. Subsequently, the reconstructed TFI is reapplied to SwinT and residual attention (RA) modulation recognition heads for refined prediction.

## A.3 ALGORITHM FLOW OF DSDM

---
**Algorithm 1** The training process of DSDM

1: **repeat**
2: $x_0 \sim q(x_0)$, $x_0 \in \mathbb{R}^{C \times L}$
3: $t = 1$, $\alpha_t = 0.5$
4: $\gamma \sim Uniform(\{1, \ldots, 10\})$ ($\gamma$ is defined in Eq:1)
5: $\epsilon \sim \mathcal{N}(\mathbf{0}, \mathbf{I})$, $\epsilon \in \mathbb{R}^{C \times D}$
6: Take gradient descent step on
$$\nabla_\theta \left\| \epsilon - \epsilon_\theta \left( \sqrt{\bar{\alpha}_t} \mathbf{x}_0 + \sqrt{1 - \bar{\alpha}_t} \times \gamma \times \epsilon, t \right) \right\|^2$$
7: **until** converged

---

---
**Algorithm 2** The inference process of DSDM

1: $t = 1$, $\alpha_t = 0.5$
2: $SIR \sim Uniform(\{1, \ldots, 10\})$
3: $x_t = \sqrt{\bar{\alpha}_t} x_0 + \sqrt{1 - \bar{\alpha}_t} \times \gamma \times \epsilon$
4: $\hat{x}_0 = x_{t-1} = \frac{1}{\sqrt{\alpha_t}} \left( \mathbf{x}_t - \frac{1 - \alpha_t}{\sqrt{1 - \bar{\alpha}_t}} \epsilon_\theta (\mathbf{x}_t, t) \right)$
5: **return** $\hat{x}_0$

---

## A.4 DETAILED INFORMATION OF DATASETS

Table 7: RML2016.10a Dataset Parameters

| Parameter Name | Parameter Values |
|---|---|
| Modulations | 8 digital modulations: BPSK, QPSK, 8PSK, QAM16, QAM64, BFSK, CPFSK, PAM4 3 analog modulations :WBFM, AM-SSB and AM-DSB |
| Number of samples per SNR per Category | 1000 |
| Length per sample | 128 |
| Signal format | In-phase and quadrature(IQ) |
| Signal dimention | 2 × 128 per sample |
| Sampling frequency | 200 kHz |
| SNR range | [-20 dB:2 dB:18 dB] |
| Total number of samples | 220,000 |

Table 8: RML2016.10b Dataset Parameters

| Parameter Name | Parameter Values |
|---|---|
| Modulations | 8 digital modulations: BPSK, QPSK, 8PSK, QAM16, QAM64, BFSK, CPFSK, PAM4 2 analog modulations : WBFM and AM-DSB |
| Number of samples per SNR per Category | 6000 |
| Length per sample | 128 |
| Signal format | In-phase and quadrature(IQ) |
| Signal dimention | 2 × 128 per sample |
| Duration per sample | 128 μs |
| Sampling frequency | 1 MHz |
| SNR range | [-20 dB:2 dB:18 dB] |
| Total number of samples | 1,200,000 |

Table 9: BT Dataset Parameters

| Dataset A(5 Gsps) | | Dataset B(10 Gsps) | | Dataset C(20 Gsps) | | Dataset D(250 Msps) | |
|---|---|---|---|---|---|---|---|
| Brand | Model | Brand | Model | Brand | Model | Brand | Model |
| Apple | iPhone 5 | Apple | iPhone 4s | Apple | iPhone 5s | Apple | iPhone 4s |
| Apple | iPhone 5s | Apple | iPhone 7 | Apple | iPhone 6s | Apple | iPhone 5 |
| Apple | iPhone 6 | Apple | iPhone 7 plus | Apple | iPhone 6s plus | Apple | iPhone 5s |
| Apple | iPhone 6s | LG | V20 | Apple | iPhone 7 | Apple | iPhone 6 |
| LG | G4 | Samsung | J7 | Huawei | Gr5 | Apple | iPhone 6s |
| Samsung | Note 3 | Samsung | Note 2 | LG | G4 | Apple | iPhone 7 |
| Samsung | S5 | Samsung | S7 Edge | Samsung | Note 3 | Apple | iPhone 7 plus |
| Sony | Xperia M5 | Xiaomi | Mi6 | Samsung | S3 | LG | G4 |
| | | | | Samsung | S3 Duos | LG | V20 |
| | | | | Samsung | S4 | Samsung | J7 |
| | | | | Sony | C4 | Samsung | Note 2 |
| | | | | | | Samsung | Note 3 |
| | | | | | | Samsung | S5 |
| | | | | | | Sony | Xperia M5 |
| | | | | | | Xiaomi | Mi6 |