# OpenReview forum: "Diffusion SigFormer for Interference Time-series Signal Recognition"
_ICLR.cc/2025/Conference — Submitted to ICLR 2025_

### Official Review · Reviewer_PhYz · 2024-10-28

**Soundness:** 1
**Presentation:** 2
**Contribution:** 1
**Rating:** 1
**Confidence:** 3

**Summary:**

This manuscript proposed an architecture that combines diffusion model and transformer to perform time-series recognition. Among them, a diffusion model called DSDM that uses SIR to constrain the noise intensity and fixed t is proposed; a 1-D convolutional layer is introduced into the Transformer to enhance the ability for extracting local features.

**Strengths:**

The authors compared the proposed method with several basic architectures, such as LSTM, ViT, and Mamba.

**Weaknesses:**

The logic of the writing itself needs to be strengthened. In the Introduction, the author describes the related research on electromagnetic signal recognition using AI and deep learning in recent years (l48-58). However, the problems mentioned in the next paragraph are not problems encountered by existing methods (l59-65). It only describes the problems encountered by traditional signal recognition. Although more recent related research is introduced in A.1 and A.2, these methods have also not been analyzed and compared with the proposed method.

The DSDM part is very confusing. In particular, the description and formula in Section 2.3 do not match the algorithm in Section A.3 at all. According to the context, algorithm in Section A.3 combines the SIR which is described in Section 2.2, and I guess Section A.3 would be a more correct description of DSDM. In addition, placing the DSDM algorithm in the related work section will make readers confused whether DSDM is the method proposed by the author or an existing method.

Adding noise to the training data is a long-established method to enhance the performance of time-series signal recognition. The experiments cannot explain whether the improvement in effect comes from the denoising process of DSDM or the noise adding for training.

Since the DSDM proposed by the author constrain the step number of diffusion processes t to 1, the author should show the relevant ablation study. Including the constrain of t, not using t as input, and comparison with the original diffusion, etc. The proposed SigFormer also needs to be verified. The author mentioned that the original transformer will cause training instability, but there is no relevant experimental comparison and analysis. Papers that use 1x1 convolution to enhance the performance of the transformer on time-series recognition have not been well cited and compared, such as [a]. And paper that add condition on signal side [b] should also be included in the comparison.

Overall, the correctness and logic of the writing of this paper need to be strengthened. There is also a lack of analytical and experimental results sufficient to verify the effectiveness of the proposed method.

[a] Eldele, Emadeldeen, et al. "Tslanet: Rethinking transformers for time series representation learning." ICML (2024).
[b] Li, Yuxin, et al. "Transformer-Modulated Diffusion Models for Probabilistic Multivariate Time Series Forecasting." ICLR (2024).

**Questions:**

Please address the problems described in weakness part.

---

### Official Review · Reviewer_rBzz · 2024-10-30

**Soundness:** 2
**Presentation:** 1
**Contribution:** 1
**Rating:** 1
**Confidence:** 4

**Summary:**

In this paper, the authors address the problem of electromagnetic signal recognition, proposing an architecture comprising multiple specialized modules. Key components include a denoising module based on a diffusion model and a transformer-based feature extraction module, SigFormer, designed for signal processing. Sevreral experiments are conducted on the RML2016 dataset to evaluate the model's performance.

**Strengths:**

The topic is interesting.

**Weaknesses:**

I strongly recommend that the authors thoroughly revise Section 2.3, beginning with clear definitions for each mathematical notation. Given that denoising diffusion models are well-studied, it is essential to differentiate prior work from the novel contributions of this paper.

I also recommend expanding Section 2.4 with a comprehensive review of existing approaches. This would help position the proposed methodology more effectively within the domain. For instance, one technique discussed involves patching a 1D signal with positional encoding. The authors are encouraged to reference existing applications of this technique—not necessarily limited to electromagnetic signal recognition—and to clarify the motivation for its use in this context.

A general comment: prior work should not be confined to the introduction or related work sections. I encourage the authors to cite relevant studies for each scientific claim and technique throughout the paper. Clearly distinguishing which methods contribute to the paper’s original work will aid readers in understanding its position within the field and will highlight its unique contributions.

Finally, a specific note: as both denoising diffusion models and transformers are well-researched topics, a straightforward application of these techniques to a new domain may not sufficiently establish originality. Greater emphasis on how these techniques are adapted or innovatively applied would strengthen the paper's contribution.

**Questions:**

N/A

---

### Official Review · Reviewer_R4Zw · 2024-10-31

**Soundness:** 2
**Presentation:** 2
**Contribution:** 2
**Rating:** 3
**Confidence:** 3

**Summary:**

The paper introduces Diffusion SigFormer, a novel approach for recognizing time-series signals with interference by combining diffusion models for denoising and transformer architectures for classification. The approach addresses challenges in recognizing electromagnetic signals in environments with interference, using a diffusion signal denoising module (DSDM) and a signal recognition model (SigFormer). The authors tested this method on multiple datasets (RML2016.10a, RML2016.10b, and a Bluetooth signal dataset), showing the feasibility  on multiple scenarios.

**Strengths:**

1. This paper proposes using a diffusion-denoising model to resolve the time-series data recognition problem. Generally, using a diffusion model to separate noise makes sense.
2. The visualization shown in Figure 4 figure 5 and Table 1 shows the feasibility of the proposed method.

**Weaknesses:**

1. The Phase Problem is not clearly explained. When applying DDPMs to time series data, one of the biggest challenges is the phase of the generated signals. However, according to Figures 4 and 5, the phase miss-match problem between the real part and the imaginary part still remains severe. However, it seems in Table 1, that this problem are not affecting a lot. How the model overcomes the phase shift and the potential impact is not clear.

2. For the experiment part Table 3, the authors compare SigFormer with Mamba and ViT.  Are the CNN-based model,  SigFormer and ViT all going with the DSDM process? Or does only the SigFormer go with the DSDM process, while others just go through with their original process for precision?

**Questions:**

1. Are the CNN-based model,  SigFormer, and ViT all going with the DSDM process?
2. Gaussian, Rayleigh, and Periodic noises seems don't have a significant difference, may I ask why authors the reason of this phenomenon?

---

### Official Review · Reviewer_ctrV · 2024-11-04

**Soundness:** 2
**Presentation:** 2
**Contribution:** 1
**Rating:** 3
**Confidence:** 5

**Summary:**

The manuscript presents an innovative approach to recognizing interference time-series signals using a novel interference signal recognition transformer. The integration of diffusion processes with transformer architectures is a novel contribution to the field of interference signal processing. Overall, the proposed method has excellent anti-interference ability.

**Strengths:**

1. The application of diffusion models with a Transformer in the context of interference signal recognition has not been fully explored. The proposed model addresses a gap in current methodologies.
2. A mechanism for electromagnetic signal interference is designed to add an appropriate amount of interference to clean signals. SigFormer combines Transformer with convolution, enabling the model to have excellent local feature extraction capabilities and perform well in global context modeling.
3. The results are well-presented, with thorough comparisons against existing models. The use of relevant benchmarks is appropriate for evaluating performance. The experimental results show the anti-interference ability of the findings.

**Weaknesses:**

1. There are several grammatical errors throughout the manuscript.

For example, in the Abstract section, the sentence "Secondly, diffusion signal denoising module was proposed to denoise the input interference signal." should be revised to "Secondly, a diffusion signal denoising module was proposed to denoise the input interference signal."

In line 45 of the introduction section, the sentence "Compared with LB methods, FB methods have stronger robustness and effectiveness, but its recognition performance depends on manually designed features and classifiers." should be revised to "Compared with LB methods, FB methods have stronger robustness and effectiveness, but their recognition performance depends on manually designed features and classifiers."

There are many similar grammatical errors in the article. The author should check them carefully.

2. Some sentences in the paper are difficult to read and understand. The author should improve his English writing skills.

For example, in the Abstract section, the sentence "We also use various types of noise to improve its denoising effect on electromagnetic signals." This sentence is not very clear. How can noise improve the denoising effect of the signal?

In line 81 of the introduction section, the sentence "In this paper, inspired by the powerful denoising ability of diffusion models and considering the interference of a large number of signals in reality, combined with the powerful temporal processing capability of transformers, we propose a novel signal recognition method based on diffusion model and transformer." The sentence is quite long and can be made clearer by breaking it up and improving some phrasing.

3. Each picture has no caption, which makes it difficult for readers to understand the picture. The method does not present enough details.

For example, in Figure 1, it is not clear how different types of noise are added to the original signals. In addition, in the Diffusion signal denoising module, does the diffusion process only go through four steps? Is the ellipsis missing?
In Figure 2, the meaning of some blocks should be explained.

4. There are some problems in the picture. In Figure 1, the image and the legend overlap. And the borderlines are very rough. In the Diffusion Process, Some arrows are missing, and the directions are not clear.

5. Some mathematical symbols are not explained. For example, "σtz" is not explained. There seems to be something wrong with the derivation from Formula 3 to Formula 4. There should be a more detailed derivation process.

6. The contributions of this paper are limited. Diffusion and transformers have been proposed for a long time. For Sigformer, transformer and convolution are just simple combinations. The method proposed in this paper seems to be a patchwork of existing techniques.

7. This paper lacks ablation experiments. Additional experiments are needed to verify the role of each component of Sigformer.

**Questions:**

1. The grammar needs to be checked for the whole paper.
2. The math notations should be explained.
3. The contribution should be highlighted.

---

> ### Comment · Reviewer_ctrV · 2024-12-02
> **Keep my original rate**
>
> I did not see any rebuttal from the authors, and I kept my original rate.

---

### Meta-Review · Area_Chair_2d4F · 2024-12-04

**Metareview:**

The paper introduces a new approach for recognizing time-series signals with interference by combining diffusion models for denoising and transformer architectures for classification. However, all the reviewers believe that the authors' English writing skills need to be improved. The figures in the paper do not have captions, and the phase problem is not clearly explained.

All reviewers voted to reject the manuscript, and the author appears to have given up. The paper is not yet ready for publication at ICLR.

**Additional Comments On Reviewer Discussion:**

The author did not engage in discussions with the reviewers, thus offering no further comments.

---

### Decision · Program_Chairs · 2025-01-22

Reject